# Current Advances in Bovine In Vitro Maturation and Embryo Production Using Different Antioxidants: A Review

**DOI:** 10.3390/jdb11030036

**Published:** 2023-08-29

**Authors:** Roksana Naspinska, Maria Helena Moreira da Silva, Fernando Moreira da Silva

**Affiliations:** Animal Reproduction, Centro de Investigação e Tecnologia Agrária dos Açores IITA-A, Faculty of Agricultural and Environmental Sciences, University of the Azores, 9701-851 Angra do Heroísmo, Portugal; roksana.naspinska@gmail.com (R.N.); helenasantossilva68@hotmail.com (M.H.M.d.S.)

**Keywords:** in vitro maturation, bovine oocytes, antioxidants, oxidative stress, embryo production

## Abstract

In vitro maturation (IVM) is one of the most important steps in in vitro embryo production (IVEP). It is a complicated procedure in which nuclear and cytoplasmatic changes in oocytes appear. In order to carry out the in vitro maturation procedure correctly, it is necessary to provide the oocytes with as close to a natural (in vivo) environment as possible. Many factors contribute to the overall poor quality of in vitro-matured oocytes. One important factor may be oxidative stress (OS). The generation of oxidants, such as reactive oxygen species, is common under culture conditions. The solution for OC treatment and prevention is antioxidants. In the last 5 years, many studies have examined different antioxidants and their effects on in vitro maturation of oocytes and embryo production. The aim of this systematic review was to present the achievements of scientific research in the last five years, in which the effects of many antioxidants were tested on bovine oocyte maturation and embryo production.

## 1. Introduction

In the last decade, many scientists have been examining the effect of antioxidants on in vitro maturation. The importance of studying antioxidants and their functions in in vitro maturation is due to their proven effect on Reactive Oxygen Species (ROS). ROS are oxygen-free radicals that damage biological systems [1]. These oxygen-free radicals decreased the success of in vitro maturation and embryo production in cattle, which is estimated at 30–40% of oocyte development into a blastocyst. Due to this result, the embryonic in vitro production efficiency is still not satisfying [2].

Antioxidants are substances that, in a small amount, occur in the organism compared to an oxidizable substrate, preventing or delaying the oxidation of the substrate. The antioxidative enzymes are the cell’s most important protection against acute oxygen toxicity [3], expressing enzymes that detoxify ROS and repair the harm they have made [4]. All aerobic organisms have evolved highly effective antioxidant strategies to prevent oxidative damage. However, these highly reactive species can be formed accidentally due to metabolic changes (e.g., during mitochondrial respiration) [5]. In vitro, the environmental parameters differ from those in vivo. For example, the oxygen concentration is higher in vitro than in an in vivo environment, leading to increased concentrations of ROS such as superoxide anion, hydrogen peroxide, and hydroxyl radical [6].

The level of ROS damage in cellular systems depends on the balance between their rate of production and removal. In the case of the overproduction of free radicals while reducing the concentration of antioxidants, a pathophysiological condition called oxidative stress (OS) occurs [6].

In vitro maturation (IVM) of oocytes is the first and most important phase of the in vitro embryo production (IVEP) process, during which oocytes gain the potential to sustain further embryonic development. As a result, identifying the right IVM environment is essential for successful IVEP procedures [7].

The success of IVM depends on various factors, including the quality of the oocytes obtained and the culture conditions [8]. One of the challenges that oocytes face during IVM is oxidative stress caused by the increased production of ROS by oocytes in an IVM medium. Excessive production of ROS may ultimately lead to oocyte death and embryo loss [9].

In this article, we will present recent studies from 2017–2023 on the effect of various antioxidants on the in vitro maturation of bovine oocytes and embryo production.

## 2. Vitamin C

Vitamin C, or L-ascorbic acid, is a water-soluble ketolactone [10]. The center of the molecule is a five-membered ring. Its breakdown leads to the formation of oxalic acid and L-threonic acid, which show strong reducing properties resulting from having endiol bonds at C2 and C3 carbons. The oxidized form of vitamin C is L-dehydroxyascorbic acid, which is also biologically active [11].

Sovernigo et al. [12] have proven that vitamin C in a quantity of 50 μg/mL added to the medium has reduced the ROS level from (36.5 ± 5.2) in the control group to (27.1 ± 3.0) in the group with vitamin C. The blastocyst rates increased from 47.2 ± 2.7% in the control group to 52.1 ± 3.1% in the group with vitamin C supplementation, and the total cell number was also higher in the embryos produced with vitamin C added to the culture media. These studies show that vitamin C may reduce oxidative stress by decreasing ROS levels.

In another article, the scientists presented the effect of ascorbic acid in the amount of 25,50,100 µM on in vitro maturation and embryonic development of buffalo oocytes. An increase in cleavage rate was observed in the groups of 25 µM (*p* < 0.01) and 50 µM (*p* < 0.001), but no effect was seen in the group with 100 µM of vitamin C compared to the control group. Another observation was that the blastocyst rate was notably increased in all groups with vitamin C supplementation, but the combination of vitamin C with cysteine improved those results even more. In the conclusion, the researchers confirmed that the best effect on in vitro embryo production has a treatment with vitamin C of 50 µM and when it is combined with 50 µM of cysteine, it shows even better results [13].

Husamaalden et al. [14] have checked the effects of vitamin C and a mix of vitamin C and cysteamine on bovine oocytes maturated in vitro, their cleavage rate, and the subsequent development of the blastocyst. Ascorbic acid in a quantity of 200 mM improved (*p* < 0.05) the maturation of oocytes and blastocyst rate after in vitro fertilization (IVF) compared to the control group but did not increase the cleavage rate. In the second experiment, to the culture medium was added cysteamine combined with vitamin C. As a result, there was no significant effect on the oocyte maturation but slightly improved (*p* < 0.05) the early cleavage (≤24 h after IVF) and 2-cell embryo formation.

## 3. Resveratrol

Resveratrol (3,5,40-trihydroxystilbene), belonging to the family of polyphenols called viniferins, has many beneficial properties, from chemoprevention to cardioprotection. Resveratrol exists in the cis or trans conformations. Although both isomers (along with their glycosides) occur in nature and appear to have similar biological properties, the effect of the trans isomer is better studied and documented [15].

The studies show that resveratrol decreased the level of ROS in bovine oocytes maturated in vitro with a result of (28.1 ± 4.7) compared to the control group (36.5 ± 5.2) while improving the in vitro maturation rate, blastocyst rate, and total number of cells in embryos [12].

Tomotaka et al. [16] have proven that resveratrol improved the development of post-warming embryos due to mitochondrial clearance, which correlated to a decrease in mitochondrial DNA copy number (Mt-number), mitochondrial protein, and dsDNA of the embryos and an increase in cell-free mitochondrial DNA in the spent culture medium of the resultant blastocysts. Resveratrol is a potent activator of SIRT1 and can increase the mitochondrial standard required to rule and maintain cellular homeostasis. In general, SIRT 1 is a nicotinamide adenosine dinucleotide (NAD)-dependent deacetylase that removes acetyl groups from various proteins, performing a wide variety of functions in biological systems [16]. Resveratrol decreased the ROS level and Mt-number of blastocysts and blastomeres without lowering the ATP content. These results show that resveratrol causes the clearance of injured mitochondria from the embryos. Summarizing, resveratrol possibly reduces Mt-number by encouraging the active removal of the broken mitochondria from the vitrified-warmed embryos, thus acting as a potent activator of SIRT1 that triggers autophagy and mitochondrial biogenesis. In another study, the researchers were investigating the influence of resveratrol and resveratrol combined with ethylene glycol tetraacetic acid (EGTA) on vitrified-warmed bovine oocytes. The experiment showed a lower level of ROS in the group with resveratrol supplementation and the lowest level of ROS in the group with resveratrol combined with EGTA. Apart from the decrease in ROS level, the addition of resveratrol improved the vitrification outcomes by increasing the development competence of oocytes compared to the ones that matured without resveratrol after warming [17].

Spricigo et al. [18] suggested the addition of L-carnitine or resveratrol to the medium for in vitro maturation of prepubertal bovine oocytes as an anti-ROS solution and to prevent high-lipid content damage. The experiment has shown that the presence of resveratrol in the culture medium decreased the percentage of oocyte apoptosis but did not have an effect on cleavage and blastocyst rates and therefore did not increase embryonic development. Another conclusion of those studies was that resveratrol combined with cysteine has a positive effect on spindle damage and gene expression.

## 4. Coenzyme Q10

Coenzyme Q10 is a compound that occurs naturally in an animal’s organism as well as in the human body. The main compounds for the synthesis of the coenzyme Q10 molecule in both prokaryotic and eukaryotic organisms are 4-hydroxybenzoate and a polyprenyl unit [19]. The main function of coenzyme Q10 comes down to its participation in the mitochondrial transport of electrons in the respiratory chain [20]. Regardless of this, coenzyme Q10 is one of the most important lipophilic antioxidants, which prevents the generation of free radicals, oxidative modifications of proteins, lipids, and DNA and contributes to the regeneration of another strong lipophilic antioxidant—a-tocopherol [21].

The purpose of the study performed by Ruiz-Conca et al. [22] was to evaluate the effects of CoQ10 on bovine oocyte in vitro maturation (IVM) and vitrification. It is confirmed that the vitrification process decreases the survival rate of oocytes. The scientists collected 311 cumulus-oocyte complexes (COCs) and cultured them with 25 μM and 50 μM CoQ10 supplementation added to the standard culture media used for IVM. After vitrification and post-vitrification warming, the oocytes were again cultured, and then their survival rate was evaluated by stereomicroscopy. The results showed that 50 μM CoQ10 supplementation improved the oocyte survival rate (57.9% vs. 77.2%; *p* = 0.045) and progression to the MII stage of meiosis (76.2% ± 3.2).

## 5. Melatonin

Melatonin is produced from endogenous tryptophan, an essential aromatic amino acid, mainly in the pineal gland, but small amounts are also produced in the retina and lens of the eye, blood cells, and the epithelium of the gastrointestinal tract [23]. It is often referred to as the sleep hormone because it mainly coordinates circadian rhythms and the rhythms of the body’s biological clock [24].

Lima et al. [25] were studying the effect of melatonin on bovine oocytes matured in heat-stress in vitro conditions in a medium with 10^−12^, 10^−9^, 10^−6^, and 10^−3^ mol/L melatonin or without in a control group. The outcomes of this study show that the best results of improved embryo quality were obtained at the quantities of 10^−6^ and 10^−4^ melatonin added to culture media.

In another study, the scientists investigated if the supplementation of the mature oocyte in the medium with melatonin could improve oocyte features and reprogramming ability. The results showed that melatonin in the amount of 10^–9^ M effectively moderated oxidative stress by decreasing ROS levels (*p* < 0.05), which decreased the apoptosis rate in oocytes. Also, it has been noticed that in the group with melatonin supplementation, there were higher proportions of oocytes with homogeneously distributed mitochondria (83.67%) than in the control group (57.61%). The abnormal spindle assembly, highly associated with meiotic arrest, was lower in the group treated with melatonin. The authors analyzed how melatonin affects epigenetic modifications in bovine oocytes. They discovered that 26 genes expressed differently in the group with or without melatonin, which can suggest the importance of those genes on correct oocyte development [26].

Yang et al. [27] made an attempt to improve the inferior oocytes (IOs), which are not capable of fertilization but often make up 1/3 or more of immature oocytes. These oocytes represent, in fact, a larger oocyte yield, exhibiting inferior ability to complete nuclear and cytoplasmic maturation in vitro due to the cumulus cells of the oocyte that protect the oocyte from the microenvironment, which helps its growth and maturation. The researchers [27] used 10^−9^ M of melatonin for culture media. In the group with melatonin (71.4 ± 1.88%), a higher rate of MII was observed compared to the group without melatonin (59.4 ± 3.14%; p < 0.05), but it was still lower than in the group of cumulus-oocyte complexes (COC) (87.9 ± 0.64%; *p* < 0.01). The ROS level in the group with melatonin was decreased compared with the no-melatonin-treated group, but the GSH level was correspondingly higher. The IOs group with melatonin supplementation had a higher level of ATP and a lower clustering distribution rate of mitochondria than the group without melatonin, and no significant difference was seen between melatonin-treated IOs and COC. The expression of genes related to oocyte maturation and embryonic development like ATPase 6, BMP-15, GDF-9, SOD-1, Gpx-4, and Bcl-2 became upregulated in the group with melatonin. All those results show a beneficial effect of melatonin on IOs and their potential use in IVF procedures.

## 6. Vitamin A

Vitamin A is a group of biologically active organic compounds. Direct precursors of group A vitamins are carotenes, which are converted into vitamins in the animal body [28]. The properties of vitamin A are demonstrated by several compounds, the most important of which is retinol (vitamin A1), as well as retinal or 3-dehydroretinol (vitamin A2). Due to their properties and chemical structure, retinoids have been divided into three generations. In the body, retinoids act through specific receptors located in the nuclei of tissues and cells, including the epidermis, hair follicles, and sebaceous glands [29] Vitamin A is involved in many processes in the human body. It plays an important role in the processes of cell differentiation and development, the regulation of proliferation, the process of vision, the growth of bone tissue, and the function of the immune system [30].

The objective of the study by Gad et al. [31] was to examine the effect of 9-cis-retinoic acid (9-cisRA) on buffalo oocyte quality and maturation rate. For the experiment, vitamin A was used in amounts of 5, 50, and 200 nM. Firstly, maturation rate and cleavage were investigated. The group with 5 nM of 9-cisRa showed the best results. The expansion and polar body rates were the highest in that group (95.8 and 45.5%, respectively), compared to the groups with another amount of 9-cisRa or the control group. On the other hand, the groups with 50 and 200 nM of 9-cisRA had the lowest maturation rates. The cleavage rate in the group with 5 nM of antioxidant was higher (61.1%) but not significantly higher compared to the control and 50 nM groups (53.8 and 57.6%, respectively). The group with 5 nM of vitamin A had higher mitochondrial membrane potential activity and a lower ROS level, and the gene expression rate was better in the group with 5 and 50 nM. The worst effect was obtained in the group with 200 nM 9-cisRA treatment.

## 7. Vitamin E

Vitamin E is the name of a group of organic chemical compounds soluble in fats that include tocopherols (T) and tocotrienols (T3). Their common feature is the presence of a bicyclic 6-hydroxychroman spine and a side chain made of three isoprene units [32]. Currently, there are eight naturally occurring homologs belonging to the vitamin E family. They are α-, β-, γ-, δ-tocopherols characterized by a saturated carbon side chain and consisting of three isoprenoid units and their equivalents in the form of unsaturated α-, β-, γ-, δ-tocotrienols [33].

In the article written by Perez [34], the results of an experiment in which 624 bovine oocytes were collected and matured in 4 groups: 0, 50, 100, and 200 mM of α-tocopherol. After maturation, the cumulus expansion index (CEI) was graded. The results showed that in the control group there was the highest maturation rate and CEI (100%; and 2.44 ± 0.20, respectively), then in the 50 mM (98.16%; 2.39 ± 0.13), 100 mM (97.40%; 2.00 ± 0.14), and 200 mM (96.25%; 2.06 ± 0.24). The results suggest that a high amount of α-tocopherol is toxic to oocytes and decreases their development rate.

In another study, two experiments were held: (1) vitrified-warmed oocytes were cultured with supplementation of 10 μM α-tocopherol and then the blastocyst formation rate was checked. Afterward, the investigation of different quantities of α-tocopherol (0, 10, 30, 100, or 300 μM) on development potential took place. As a result, there was a significant difference in day 8 blastocyst yield, between the control and supplemented groups (25.8% vs. 36.3%), but no effect was observed on the day 2 cleavage rate. (2) Vitrified-warmed oocytes supplemented with α-tocopherol were investigated in the parameters of ROS level, mitochondrial activity, and distribution of cortical granules. There was no significant effect observed in those parameters, but the percentage of zygotes exhibiting normal single aster formation was improved compared to untreated oocytes (90.3% vs. 48.0%) [35].

The study by Singh et al. [36] determined the effect of Epigallocatechin Gallate (EGCG), Alpha-tocopherol, and their combination on IVM and IVF parameters. EGCG is a type of plant-based compound called catechin, which acts as a potent antioxidant that may protect against cellular damage caused by free radicals.

At a concentration of 10 μM, EGCG increased cumulus expansion and 1st and 2nd polar body extrusion (*p* < 0.05) compared to the control group and other groups with antioxidants. Supplementation of 100 μM of alpha-tocopherol has not improved IVM and IVF rates. The combination of 10 μM EGCG and 100 μM alpha-tocopherol increased (*p* < 0.05) IVM and IVF parameters as well.

## 8. Carotenoids

Carotenoids are substances that give yellow to red colors to both plants and animals. They can be divided into carotenes (C40 hydrocarbons, e.g.,(ψ,ψ-carotene, Lyc), β-Apo-8′-carotenal (β-ACA), torulene (3′,4′-didehydro-β,ψ-caroteneTor), isorenieraten, and their oxygen derivatives, xanthophylls (containing oxygen in the molecule in the form of hydroxyl, epoxide, or carbonyl groups, e.g., zeaxanthin, lutein, astaxanthin, canthaxanthin, echinenone, 3,3′ dimethoxyisorenieraten [37]. Carotenoids that are composed of the 11 conjugated double bonds can be included in the group of polyisoprenoids and are low-polar substances. They can occur in acyclic, monocyclic, or bicyclic forms [38]. Unlike plants, animals cannot synthesize carotenoids on their own as a result of biochemical processes; therefore, these substances must be supplied to the body through the diet [39].

The supplementation of 0.2 μM lycopene to the oocyte in vitro maturation and embryo culture media has observed a significant (*p* < 0.05) improvement in cleavage and blastocyst development rates compared to the controls (84.3 ± 0.6% vs. 73.1 ± 1.9% and 41.0 ± 1.4% vs. 33.4 ± 0.7%, respectively). Lycopene also reduced (*p* < 0.05) intracellular ROS concentrations in oocytes and blastocysts. In the group with lycopene supplementation, the apoptosis rate of the produced embryos was lower and the total cell number was higher than in the control. IκBKB (Inhibitor of nuclear factor kappa B kinase, subunit beta), Caspase 9, and Caspase 3 were down-regulated in the group with lycopene, and on the other hand, GDF9 (Growth and differentiation factor 9) and BMP15 (Bone morphogenetic protein 15) were significantly up-regulated in the same group. The conclusion of this study was that supplementation with lycopene has a positive effect on improving embryonic resilience to stress [40].

The goal of another study was to check the effect of increasing dissolved oxygen availability using a gas permeable (GP) culture device with or without astaxanthin (Ax). The supplementation occurred on 8-day IVG culture systems for bovine OCGCs. The oocytes were retrieved from early antral follicles and then cultured in GP, GP with Ax (GP + Ax), and a conventional gas-impermeable device (control) for 8 or 12 days. Then the researchers check the parameters of oocytes: viability rate, granulosa cell number, oocyte diameter, nuclear maturation rate, ROS generation, cleavage, and blastocyst rates. The results showed that in the groups with GP and GP + Ax, no significant change was seen in viability rate, granulosa cell number, or oocyte diameter compared to the control group. On day 8, the nuclear maturation rate was higher in the GP + Ax group (*p* < 0.05) than in the GP and control groups, but on day 12, the results were similar for all groups. The ROS level was lower in the group with GP + Ax, which shows a positive effect of antioxidants on reducing oxidative stress. The cleavage and blastocyst rates did not vary in each group. It is suggested that supplementation with astaxanthin may improve oocyte maturation after 8 days of culturing [41].

## 9. Thiols

The antioxidant properties of thiol compounds are revealed through various mechanisms. These compounds are a component of the thiol-disulfide redox buffer; they are free radical scavengers as well as metal ion chelators. Thiols inhibit the oxidation of low-density lipoproteins (LDL) in human plasma [42]. Thiol compounds, such as glutathione (GSH), may be substrates in specific redox reactions and participate in the reduction of disulfide bridges in proteins, and GSH biosynthesis in cells decreases with age [43]. Protein thiols undergo S-thiolation reactions both with GSSG (S-S-G protein) and with cysteine (S-S-cys protein), which leads to the formation of mixed disulfides [44].

De Mattos et al. [45] investigated the effects of β-mercaptoethanol (βME) supplementation to the culture medium of bovine in vitro-produced (IVP) embryos before or after vitrification on embryo development and cryotolerance. There were two experiments held. In the first experiment, the blastocysts from Day-7 IVP were cultured in a medium inclusive of 0, 50, or 100 μM βME for 72 h. In the second experiment, IVP embryos were cultured to the blastocyst stage in 0 (control) or 100 μM βME and then vitrificated and warmed. The results showed that supplementation with βME during in vitro maturation reduced embryo development (28.0% vs. 43.8%) but, after vitrification, improved the re-expansion rate, so it can be said that βME increases cryotolerance of vitrified embryos.

In another article, the effects of β-mercaptoethanol (βME) supplementation on early apoptosis, intracellular GSH contents, oocyte maturation rate, cleavage, and embryonic development rates after in vitro fertilization (IVF) have been investigated. In the experiment, β-mercaptoethanol was used in quantities of 50, 100, 150, and 200 μM, and oocytes were divided into 3 groups depending on their cumulus cell structure. The groups were called A, B, and C. Group A contained compact and dense cumulus cells; group B contained compact but less dense cumulus cells; and group C contained a thin or small remnant of cumulus cells. The results of the study show that supplementation of βME to the culture media during in vitro maturation increased GSH synthesis and early apoptosis in groups B and C and decreased in group A. The addition of 50 μM BME improved the maturation rate of group A but had no effect on oocyte development competency. It was noticed that a concentration higher than 100 μM is harmful to early embryonic development [46].

Sandal et al. [47] suggested that the production of GSH depends on the supplementation of cysteine in the medium. The cysteine is very fugitive outside the cell and is auto-oxidized to cystine. The GSH synthesis is enhanced by cysteine due to its metabolic changes, which are promoted by cysteamine. To summarize, cysteamine plays an important role in the synthesis of GSH, which plays a part in the defense mechanism against ROS.

Table 1 represents the summaries of researches carried out in recent years on the effect of different antioxidants on the rate of maturation, cleavage and production of blastocysts, discussed in this review paper.

## 10. Conclusions

In recent years, many scientists have conducted studies investigating the effect of antioxidants on reducing the reactivity of free radicals in order to protect oocytes from oxidative stress. Most of these studies have confirmed that antioxidants reduce the number of ROS, improve the parameters of oocytes, and have a positive effect on the developmental competence of embryos. The most recent results obtained by different authors and presented in this review evidently show how the use of antioxidants in the field of embryo production increases the effectiveness of in vitro maturation and further embryonic development after in vitro fertilization.

## Figures and Tables

**Table 1 jdb-11-00036-t001:** Effect of different antioxidants on the rate of maturation, cleavage and production of blastocysts.

Antioxidant	Maturation Rate		Cleavage Rate		Blastocyst Rate		Reference
	Control	With Antioxidant	Control	With Antioxidant	Control	With Antioxidant	
Resveratrol	86.4 ± 2.7	91.8 ± 3.0	85.9 ± 4.1	88.7 ± 8.4	47.2 ± 2.7	54.2 ± 4.0	[12]
Coenzyme Q10	66.0 ± 10.6	76.4 ± 12.2	67.9 ± 9.8	74.4 ± 9.2	43.5 ± 7.1	45.1 ± 7.4	[48]
Melatonin	81.2 ± 4.2	92.8 ± 2.3	53.1 ± 2.4	62.1 ± 2.8	14.1 ± 2.4	19.1 ± 1.1	[49]
Vitamin A	--	--	66.7 ± 2.7	70.1 ± 3.9	21.9 ± 1.9	24.2 ± 2.7	[50]
Vitamin E	81.1 ± 1.8	94.5 ± 5.2	54.9 ± 1.2	60.2 ± 4.1	14.6 ± 19.1	19.1 ± 1.9	[49]
Vitamin C	80.2 ± 2.8	85.0 ± 2.3	56.5 ± 4.2	65.3 ± 3.1	12.3 ± 1.6	20.2 ± 1.1	[49]
Carotenoids	66.3 ± 3.7	76.0 ± 1.9	54.2 ± 2.2	68.8 ± 2.0	28.0 ± 1.2	35.2 ± 1.5	[40]
Thiols	--	--	61.8 ± 5.4	71.1 ± 5.2	15.7 ± 5.7	23.8 ± 5.9	[12]

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
