# Peer review of "Current Advances in Bovine In Vitro Maturation and Embryo Production Using Different Antioxidants: A Review"

_jdb, 2023, doi:10.3390/jdb11030036_

Round 1

Reviewer 1 Report

This review describes the influence of adding various antioxidants to culture media on cattle oocyte and embryo quality.  

Some English grammar improvement is warranted throughout.

Inclusion of a short review of currently used and most productive IVM systems in cattle is warranted

Discussion of IVF vs IVM vs IVEP needs to be more clearly defined and organized by section.    Are the antioxidants being discussed always only add to IVM medium?  Are they ever added to the IVF and/or IVC medium?  this is not clear 

For each section, there is lack of data on oocyte origin.  Are all oocytes discussed of slaughterhouse origin?  If not please include details.  Similarly, there is no inclusion of the base media used.  The presentation suggests that each study started with the same base media and only the antioxidants included were varied.  This seems unlikely, and the base media can influence overall results so this should be included as much as possible.  

Vit C

Line 62 – Which ROS specifically?  Added to the IVM medium only?  What about the IVF and IVEP media?

Throughout, the authors refer to improved blastocyst production, but do not provide specific percent values and/or differences in production. 

Resveratrol

Line 98 – SIRT1 needs to be defined the first time it is used

Melatonin

Line 159 – please define what an inferior oocyte is?  Does this refer to oocyte quality grade?   Cumulus amount and/or cytoplasm appearance?  

Line 163 – do the authors suggest that the oocytes that were cumulus oocyte complexes were the only ones that were ‘capable of fertilization?  Please clarify this comparison

Vit E

Line 222 – what is EGCG?

Carotenoids

Line 244 – apoptosis of what?   Please clarify 

Line 250 – the authors introduce the concept of varying gas culture environments.  This adds a new level of complexity to the discussion as the percent of oxygen that each system is exposed to will change the reaction to the antioxidants added.  Since this is not discussed in other sections, suggest removing it from this section. 

Conclusion

Line 307 – The improvement of oocyte ‘quality’ as measured by many metrics does not only improve the results of IVF but of all aspects of the entire embryo production system.  This system is multi-faceted and complex.  The overall results are improvement not only in production at each stage of the system but also in monetary expense, and staff (time and effort) expense.  A mention of this is warranted.   

Overall

The structure of the manuscript is disjointed.  The authors discuss many endpoints/metrics of development and quality (embryo cell number, spindle formation, vitrification survival, day8 blastocyst yield, damaged mitochondria, timing of first cleavage) as well as measurements of ROS in media.  However, there is no order to or consistency in the presentation of these metrics.  There is also a notable lack of clarity on metrics specific to each stage of development (IVM IVF IVEP). It would be helpful to the reader if the structure of each section was in the same order of discussion (either by metric or stage of culture being discussed).  As it is written, it is impossible to draw conclusions across and among the different antioxidants discussed.  Inclusion of a summary table would be immensely helpful. 

Author Response

Dear Reviewer,

In the attachment, I am sending the answers according to the article. 

Kind regards,

Roksana NaspiÅ„ska 

Reviewer 2 Report

This systematic review aimed to present the achievements of scientific research in the last five years, in which the effects of some antioxidants on bovine oocyte in vitro maturation and embryonic production. This is a valuable topic. It could be considered for publication after revision.

1. Give more information about the ratio of IVM.

2. Make a table to for these antioxidants, including the concentration, increased ratio compared with the control, possible mechanism, references, etc.

3. Guess which antioxidant works best.

Author Response

Dear Reviewer,

In the attachment, I  am sending the answers according to the article,

Kind regards,

Roksana NaspiÅ„ska 

Reviewer 3 Report

The review article entitled “Current advances in bovine in vitro maturation and embryo production using different antioxidants” aimed to present the achievements of scientific research in the last five years in which the effects of many antioxidants were tested on bovine oocyte maturation and embryo production. The review was written well, however, I have identified some major issues that should improve the quality of the review from my point of view: 

- There are several compounds that have antioxidant effects in bovine oocyte maturation and are not included in this review such as quercetin, alpha lipoic acid, zinc, selenium, green tea catechins, Syzygium aromaticum, Thymus vulgaris L, etc. 

- I suggest rearranging the review under the main titles such as Vitamins (subtitles: Vitamin A, vitamin E, Vitamin C), Polyphenols (subtitles: resveratrol, Quercetin, etc.), minerals (subtitles: zinc, selenium), etc.

- I suggest adding a title after the introduction about sources of oxidative stress in in vitro embryo production and their effect on oocyte maturation and development competence and supporting it with figures.

- I suggest summarizing the important data in a table.

Author Response

(The authors gave the same response as above.)

Round 2

Reviewer 3 Report

Regrettably, the authors have not implemented the majority of the suggested improvements aimed at enhancing the quality of the manuscript.